# Personhood Beliefs in Dementia Care: Influences of Race, Socioeconomic Factors, and Social Vulnerability

**DOI:** 10.3390/ijerph22101491

**Published:** 2025-09-26

**Authors:** Taniya J. Koswatta, Samantha Hoeper, Peter S. Reed, Jennifer Carson

**Affiliations:** 1Sanford Center for Aging, School of Medicine, University of Nevada, Reno, NV 89557, USA; psreed@unr.edu; 2Department of Health Behavior, Policy and Administration Sciences, School of Public Health, University of Nevada, Reno, NV 89557, USA; shoeper@unr.edu (S.H.); jennifercarson@unr.edu (J.C.); 3Dementia Engagement, Education, and Research (DEER) Program, School of Public Health, University of Nevada, Reno, NV 89557, USA

**Keywords:** personhood beliefs, dementia, race, age, social vulnerability

## Abstract

Beliefs about personhood held by healthcare professionals and care partners influence care outcomes, satisfaction, and the well-being of persons living with dementia (PLWD). This study examined differences in personhood beliefs based on demographic and contextual factors, including the Social Vulnerability Index (SVI), using registration data from the *Bravo Zulu* care partner training program (*n* = 540). Guided by the Ring Theory of Personhood, eight factors were analyzed: age, sex, race, socioeconomic status, professional discipline, healthcare experience, prior care partner training, and SVI. One-way ANOVA and independent *t*-tests were used to examine group-level differences, and multiple linear regression was conducted to assess the extent to which these factors predicted personhood beliefs. Race, age (borderline significance) professional discipline, and prior training as a care partner were significant predictors of personhood beliefs. Subscale analyses using ANOVA and *t*-test showed that beliefs about psychosocial engagement varied by SVI and healthcare experience with small effect size; however, these factors did not significantly predict of overall personhood beliefs in the regression model. Findings underscore the importance of recognizing how background characteristics shape personhood beliefs about PLWD. Promoting self-reflection and expanding culturally responsive training may support person- and relationship-centered care and improve satisfaction in multicultural care settings.

## 1. Introduction

Personhood in healthcare, dementia care, and aging services refers to recognizing and respecting the individuality, dignity, autonomy, and inherent value of each person receiving care. It is a foundational principle of person-centered care, which is widely regarded as essential to improving quality care and outcomes [1]. Personhood is also central to relationship-centered care, which builds on the foundation of person-centered practices [2] by expanding the emphasis on the role of reciprocity and belief that optimal care and support can only be achieved when all parties involved in the context of care experience well-being [3]. Honoring personhood not only enhances satisfaction for both care recipients and their families [4], but has also been linked to reduced healthcare utilization and lower overall costs through decreased reliance on specialty care services [5]. According to the Knowledge, Attitude, and Practice (KAP) theory, sustainable changes in clinical practice begin with building knowledge, followed by the development of positive attitudes, which in turn influence behavior change [6,7,8]. When applying KAP to achieving person-and relationship-centered care, healthcare professionals, along with family care partners, must first develop an understanding of person-centered principles and hold supportive attitudes toward such principles as a precursor to positively modifying their care practices. Therefore, healthcare providers’ beliefs about personhood are a key element to achieving person- and relationship-centered care and promoting higher satisfaction with the care received [4]. Notably, close emotional connections between care partners and persons living with dementia (PLWD), as well as meaningful professional relationships, play a vital role in sustaining personhood [9].

Personhood beliefs are especially important in care of PLWD, as the progression of dementia often impairs communication and memory, challenging how individuals are perceived and supported by others [10,11]. Kitwood’s conceptualization of personhood as socially constructed acknowledges the essential attributes of a person, including their unique history, experiences, and connection between individuals [12]. As dementia advances, a person’s ability to recall their personal history and experiences becomes increasingly limited [13], which may lead to a loss of recognition and respect from healthcare professionals or other care partners [14]. Past research has shown that the manner in which healthcare professionals or care partners perceive the personhood of PLWD affects both the quality of care provided as well as the well-being of both care recipients and care partners [12,15,16].

The Ring Theory of Personhood (RToP) expands on this understanding by outlining four interconnected dimensions: innate, individual, relational, and societal [17]. This theory highlights that personhood is not only an inherent and individual attribute, but that it is shaped, sustained, or diminished through social relationships and broader societal influences [18]. In care settings, interactions between healthcare professionals and care recipients are not merely transactional, but rather they involve mutual recognition, empathy, and shared meaning [19,20,21]. Personhood beliefs, therefore, are informed not only by individual values, but also by interpersonal relationships, cultural norms, and institutional expectations [22,23]. These beliefs may be further shaped by cross-cultural differences, particularly in multicultural care settings where values and expectations around providing care and support can vary. Prior studies have shown that a care partner’s cultural background, professional role, and life experiences influence their perceptions of personhood, which in turn affect the nature and outcomes of care [24,25,26,27,28].

According to the 2020 Census, the U.S. population has become increasingly racially and ethnically diverse, with the diversity index rising from 54.9% in 2010 to 61.1% in 2020 [29]. This means that if two people are selected at random, there is a 61.1% chance that they will belong to different racial or ethnic groups [29]. As of 2023, population estimates show that 58% of the U.S. population is White (non-Hispanic), 19% Hispanic, 14% Black or African American, and 6% Asian [30]. The healthcare workforce is also becoming more diverse, though some professions still do not fully reflect the diversity of the broader U.S population [31]. According to the National Center for Health Workforce Analysis (2024), the physician workforce is 62% White, 22% Asian, 7% Hispanic or Latino, and 5% Black or African American, while the nursing workforce is 65% White, 13.8% Black, 9.2% Hispanic, and 8.5% Asian [31]. Given this increasing diversity in both the aging population and the healthcare workforce, it is critical to understand how factors such as race, socioeconomic status, and professional background influence personhood beliefs toward PLWD.

The present study included two primary objectives: (1) To determine whether personhood beliefs held by healthcare professionals and other care partners toward PLWD differ based on demographic and contextual factors (e.g., race, socioeconomic status, healthcare experience, and professional discipline); and (2) To assess the extent to which these factors predict overall personhood beliefs among healthcare professionals and other care partners. These objectives were examined using secondary data (i.e., registration data) from a care partner training program named *Bravo Zulu: Achieving Excellence in Relationships-Centered Dementia Care* [32]. The training aims to deepen understanding of personhood, relationships, and culture to support compassionate, respectful, and personalized dementia care.

### 1.1. Conceptual Framework

To guide this analysis, we drew on the RToP, which provides a holistic framework for understanding how a range of individual, relational, and societal factors collectively shape beliefs about personhood [17]. According to the RToP, there are four rings: first, the ‘innate ring,’ which represents the core of personhood, encompassing fundamental human characteristics tied to biological life; second, the ‘individual ring’ includes personal values, beliefs, and personality traits that highlight each individual’s unique identity; third, the ‘relational ring’ addresses the impact of social relationships, and specifically close reciprocal connections; and fourth, the ‘societal ring’ reflects how the broader societal beliefs, values, norms, and standards influence personhood attitudes [17,18].

Using this framework and available secondary data, we examined eight factors hypothesized to influence personhood beliefs: age, sex, race, socioeconomic status, healthcare experience, professional discipline, prior training as a care partner, and community-level social vulnerability (SVI). Variable selection was guided by the RToP and aligned with the data available in the secondary dataset. The following sections review existing literature supporting the relevance of each factor.

### 1.2. Age

As individuals age, their understanding of themselves evolves, integrating past experiences, relationships, and shifting values [17,33]. RToP identifies age as a key component within the innate domain, while emphasizing that relational factors, such as family and community connections, gain greater importance as people age [34]. In the context of dementia, age is particularly relevant to personhood beliefs. Individuals with young-onset dementia often face unique challenges in fitting into typical dementia care environments, which are primarily designed for those aged 65 and older [33]. Their social expectations, shaped by age and lived experience, differ significantly from those of older adults with dementia [33]. Older adults were more likely to feel comfortable around individuals living with dementia compared to younger and middle-aged individuals [35]. Furthermore, age-related changes impact healthcare decision-making and care preferences in general care settings. A population-based study found that people tend to shift preferences and deference from patient-directed care to physician-directed care around the age of 45, meaning that the older one gets, the more likely they are to defer to their physician’s care decisions [36]. Additionally, older physicians tend to develop stronger patient–physician relationships than their younger counterparts, further demonstrating the relational influence of age on personhood beliefs [21].

### 1.3. Sex

Attitudes toward person-centered care appear to differ based on sex or gender, although many studies do not clearly differentiate between biological categories (e.g., male, female) and social identities (e.g., man, woman). Female healthcare professionals and health professions students have been found to be more responsive to information about personhood and dignity than their male counterparts [4]. Similarly, female healthcare students reported significantly higher attitudes toward patient-centered care, which shares core principles with person-centered care, such as respect for and responsiveness to individual needs and values [37,38]. Additionally, in healthcare decision-making, female care recipients tend to prefer person-centered and collaborative approaches compared to males [36,39]. In some cases, concordance between a provider’s and a patient’s sex or gender has been associated with greater adherence to person-centered care practices [40]. However, in the specific context of dementia, prior research has not found significant differences in personhood beliefs based on sex or gender [32,35].

### 1.4. SVI and Socioeconomic Status

The concept of personhood is a dynamic construct that is influenced by various psychological, social, and cultural factors. Within RToP, the societal ring represents the broader societal structures, roles, responsibilities, and expectations that individuals are bound to by virtue of being part of a specific community [17,18]. This includes societal norms, legal and ethical obligations, and collective values that influence how individuals are treated and how they behave [18]. The SVI is a community-level metric incorporating 16 indicators such as income, education, housing, and access to transportation [41]. SVI provides insights into how non-health-related dimensions affect individuals’ recovery from health events [42]. For instance, 75% of health outcome studies identified SVI as a significant predictor of health outcomes [42]. Patients with higher SVI scores were shown to have worse postoperative outcomes [43]. Because SVI reflects the broader social context including economic status and societal expectations it aligns with the societal ring of the RToP. This provides a way to quantify how community-level vulnerabilities shape individual perceptions of personhood and influence health-related beliefs and behaviors.

### 1.5. Race

Cultural perspectives on personhood vary significantly across Western and non-Western cultures [44]. While Western cultures emphasize individualism, focusing on autonomy and personal rights, non-Western cultures, such as those in Asia and Africa, have been noted as generally adopting a relational perspective, emphasizing family roles, community ties, and harmony [26,45,46]. These cultural differences are reflected in healthcare practices, with Western approaches prioritizing autonomy, and traditional Asian contexts emphasizing relational care [24,26]. Studies have shown variability in person-centered care across racial and ethnic groups, with racial concordance positively influencing communication, care satisfaction, and respect [47,48]. Additionally, care partners’ attitudes toward personhood differ across cultures; for example, Korean care partners report lower personhood beliefs for PLWD compared to Canadian care partners [27].

### 1.6. Professional Discipline and Healthcare Experience

Research demonstrates variability in personhood perceptions across healthcare disciplines, such that nursing and medical students tend to score highest in attitudes toward personhood, followed by social workers, chaplains, healthcare aides, nurses, physician residents, and physicians, who score the lowest [4]. In the context of dementia care training, these disciplinary differences in personhood perceptions indicate the potential for variation both in the prior disciplinary views brought into the training and in the change in perceptions resulting from the training. Similarly, the Person-Centered Practice Framework emphasizes the importance of healthcare professionals’ attributes, such as professional competence, interpersonal skills, job commitment, clarity of beliefs and values, and self-awareness, in fostering effective person-centered care [49]. For example, experience plays a critical role, with healthcare professionals having either no experience or over 15 years of experience demonstrating higher personhood scores than those with between 0 and 15 years of experience [4].

Guided by the RToP and the supporting literature reviewed above, we hypothesized that personhood beliefs would differ significantly based on key demographic and contextual factors, including age, sex, race, socioeconomic status, professional discipline, healthcare experience, prior care partner training, and SVI. While prior research has explored personhood in specific healthcare roles or settings, such as intensive care units [18], senior nurses [50], or palliative care physicians [51], no research, to our knowledge, has explored multiple factors collectively across a diverse group of healthcare professionals. In particular, the connection between community-level social vulnerability (i.e., SVI) and personhood beliefs remains largely unexplored. Furthermore, there is a lack of comparative studies in multicultural contexts, such as racially diverse populations like our sample. This study addresses these gaps by analyzing secondary data collected through workshop registration from a dementia care training program. It provides insight into how demographic and contextual factors jointly influence personhood beliefs and offers a foundation for future, more detailed research.

## 2. Materials and Method

### 2.1. Study Population

Data for this study were drawn from a program evaluation of the *Bravo Zulu: Achieving Excellence in Relationship-Centered Dementia Care* training program which was designed to enhance relationship-centered care and cultural competence among both healthcare professionals and non-professional care partners. The training is free and open to any individual interested in dementia care. Participants who completed the training were eligible to receive 12.0 Continuing Education Units (CEUs), approved for Long-Term Care Administrators, Marriage and Family Therapists, and Social Work professionals in Nevada. Additional information on the training structure is available in Carson et al. [32]. The training sessions were conducted from March 2020 to May 2024. Upon registration, all participants were asked to complete the Personhood in Dementia Questionnaire (PDQ) by Hunter et al. [52]. IRB approval for secondary data analysis was obtained (IRB No. 2258703-1).

To ensure data integrity and consistency in score interpretation, only participants who completed all 20 items of the PDQ were included in the analysis. Given the adequate sample size with complete data (*n* = 540), cases with missing data were excluded rather than imputed, as the PDQ yields a total score representing latent beliefs about personhood, and imputing or substituting missing responses risks compromising the validity of this score. Additionally, to avoid any potential testing effects, repeat training attendees were not considered eligible participants; only the registration data from their first participation cycle were included as a valid response. This approach ensured that repeated trainees, who might have developed a more refined perspective on the personhood approach due to their prior participation in *Bravo Zulu* training, did not skew the results. A total of 540 participants met the inclusion criteria. Approximately 80% of the sample resided in Nevada, with the remaining 20% residing in other U.S. states. Thus, while the results in this study provide many key insights about the role participant background play in personhood beliefs, there are limits to the study’s full generalizability to State of Nevada or United States. All participants were asked to provide their residential ZIP code upon registration. Using the ZIP code, we identified the corresponding county and state. Additionally, we incorporated the county-specific SVI value using the 2022 overall SVI nationwide comparison data [53]. Table 1 provides the demographic information of the study sample.

### 2.2. Dependent Variable: Personhood Beliefs

Personhood beliefs were measured using the PDQ instrument developed by Hunter et al. [52]. The PDQ is a validated instrument with demonstrated internal consistency (Cronbach’s α = 0.81) [52] and has been translated into Korean [27] and Italian [54]. It has been widely used to assess beliefs about the personhood of PLWD among healthcare professionals including long-term care staff. For this study, minor adaptations were made to align the instrument with the target population. Specifically, the phrase “resident with dementia” was replaced with “people living with dementia” to reflect the broader audience of the *Bravo Zulu* training program, which includes both professional and non-professional care partners, both within and external to long-term care settings. Additionally, the original 7-point Likert scale was modified to a 5-point scale (5 = strongly agree to 1 = strongly disagree) to reduce respondent burden in the context of program evaluation. Prior research has shown that 5-point and 7-point Likert scales yield comparable results [55], supporting the appropriateness of this modification. Further, analyses within this study demonstrated high internal consistency for the modified PDQ (α = 0.90) illustrating the reliability of the modified instrument. Items 4, 5, 8, 14, 16, 17, and 18 were reverse coded as per the original scoring guidelines.

To further explore the structure of personhood beliefs, we applied the three-factor subscale model proposed by Kim et al. [27], which includes

*Psychosocial engagement*: perceptions of PLWD’s capacity for psychological and social connection (items 9, 10, 11, 12, 13, 14, 15, 19, and 20).*Respect for personhood*: beliefs about the moral status and inherent worth of PLWD (items 4, 5, 7, 8, 16, 17, and 18).*Agency*: beliefs about PLWD’s ability to make decisions and maintain autonomy (items 1, 2, 3, and 6) [27] (p. 8).

These subscales were chosen for three key reasons: (1) they are grounded in the theoretical foundations of the original PDQ [27]; (2) they demonstrated good reliability in prior work (Cronbach’s α = 0.76, 0.77, and 0.82, respectively) [27]; and (3) their use was supported through personal communication with the original instrument developer (P.V. Hunter, 16 February 2024). To ensure the subscales were appropriate for our sample, we conducted confirmatory factor analyses in R using the lavaan package [56]. We tested the three-factor structure proposed by Kim et al. [27]. The Kim’s [27] model showed reasonable fit (CFI = 0.84, TLI = 0.81, RMSEA = 0.09, SRMR = 0.08), with most items loading on their expected factors. While the Kim’s [27] model built on our dataset did not perfectly fit, it provided the best balance between theoretical alignment and empirical performance. It also outperformed alternative structures [54], making it the most appropriate model for analyzing the PDQ in this study. At the subscale level, internal consistency was strong, with Cronbach’s alpha values of 0.78 for *agency*, 0.83 for *respect for personhood*, and 0.85 for *psychosocial engagement*, indicating good reliability across subcomponents. These findings suggest that both the overall scale and its subscales were reliable in the context of our study. Based on these results, and consistent with the recommendation of the original PDQ author, we retained the Kim three-factor structure for all subsequent analyses, while acknowledging its limitations.

In the current sample, PDQ scores ranged from 60 to 100, with a mean of 87.62 (*SD* = 9.36), a 5% trimmed mean of 88.18, a slight negative skew (−0.66), and kurtosis of (−0.082). Two cases (0.37%) were identified as outliers based on the overall distribution, with standardized residuals exceeding ±3, and were removed prior to conducting inferential statistical analyses to maintain the integrity of the results, yielding a final analytic sample of 540.

### 2.3. Independent Variables

We classified demographic and contextual variables as follows:Sex: self-reported as male or female;Race: categorized as White, Black, Hispanic, or Asian; Participants who selected multiple races or belonged to a group with fewer than 10 respondents were excluded from the ANOVA to minimize noise and ensure more comparable group sizes;Age: calculated from birth year and grouped into 18–34, 35–44, 45–54, 55–64, and 65+ categories;Socioeconomic status: based on self-report of receiving public assistance (yes/no);Formal care partner training: (yes/no) (As previously noted, the authors’ preferred terminology is ‘care partner’ rather than ‘caregiver’, and thus while the survey item referred to ‘caregivers’ and ‘caregiving training,’ these terms have been changed to ‘care partner’ and ‘care partner training’ for the purposes of this manuscript);Healthcare professional experience: yes/no (defined as having worked as a physician, nurse, nurse’s aide, or direct care worker);Professional discipline: self-identified from seven categories; medicine, nursing, and dentistry combined for analysis; andSVI: county-level CDC 2022 percentile rank, classified into four levels following the 4-level CDC classification: low (0.0–0.2500), mid-low (0.2501–0.5000), mid-high (0.5001–0.7500), and high (0.7501–1.0000).

### 2.4. Data Analysis

All analyses were conducted using SPSS version 29 (IBM Corp., Armonk, NY, USA). Descriptive statistics were used to summarize participant characteristics as well as total and subscale scores on the PDQ. To examine differences in personhood beliefs across demographic and contextual variables, we first conducted group-level comparisons using one-way ANOVA for predictors with more than two categories (e.g., race, age group, SVI category, and professional discipline) and independent *t*-tests for binary variables (e.g., prior care partner training, healthcare experience, socioeconomic status, and sex). These bivariate analyses were performed to identify patterns of group differences and inform the interpretation of multivariate results. Effect sizes were evaluated using partial eta-squared (*η*^2^) for ANOVA and Cohen’s d for *t*-tests, following thresholds for small, medium, and large effects [57].

To assess the predictive value of demographic and contextual factors on overall personhood beliefs, we performed a multiple linear regression analysis using the total PDQ score as the dependent variable. While Age and SVI were initially introduced as categorical variables for descriptive and group-level analyses (e.g., ANOVA, *t*-tests), they were treated as continuous variables in regression models to preserve the full range of variability and improve statistical power. This multivariate approach enabled us to examine the unique and combined influence of all factors while accounting for potential confounding relationships. Additionally, we conducted subscale-level analyses based on the three-factor structure *(agency, respect for personhood, and psychosocial engagement*) proposed by Kim et al. [27] to explore whether specific domains of personhood beliefs varied by participant characteristics. This allowed for a more nuanced understanding of which specific dimensions of personhood beliefs are most sensitive to demographic and contextual variation.

## 3. Results

The study sample included a diverse group of participants in the *Bravo Zulu* training program, with 62.5% of participants reporting healthcare professional experience and 37.5% reporting none. ANOVA and *t*-tests on total personhood beliefs towards PLWD showed mean score differences based on participants’ race, SVI, age, professional discipline, and prior formal care partner training. Participants who self-identified as White or Black scored, on average, five points higher in personhood beliefs compared to those identifying as Asian or Hispanic. Individuals living in counties classified as having low or low-medium SVI scores reported higher personhood beliefs than those in counties with medium or high SVI scores. A clear trend was observed based on the age of participants: mean personhood scores were relatively stable across the 18–34, 35–44, and 45–54 age groups (all around 87), increased slightly in the 55–64 group (mean = 89), but declined in the 65+ group (mean = 83).

Professional discipline also influenced personhood beliefs. Participants working in public health or ‘other’ health professions reported higher scores than those working in identified health professional categories. Additionally, participants who had received prior formal care partner training scored higher on personhood beliefs compared to those without such training. No statistically significant differences in total personhood scores were observed based on sex, socioeconomic status, or prior healthcare professional experience (See Table 2).

At the subscale level, beliefs about *psychosocial engagement* varied across all independent variables except sex and socioeconomic status (see Table 3). For the *respect for personhood* subscale, mean scores differed by race, age, and prior formal care partner training. In the *agency* subscale, differences were observed based on professional discipline and prior formal care partner training. Notably, prior formal care partner training showed significant differences across all three subscales, with those having prior training reporting higher *personhood beliefs*.

Confirming the ANOVA and *t*-test findings, a multiple regression analysis resulted in a significant model, *F*(13, 382) = 5.31, *p* < 0.001, *ƒ*^2^ = 0.18 (medium effect) and explained approximately 15.3% (*R*^2^ = 0.153) of the variance in personhood beliefs (See Table 4). A post hoc power analysis using G*Power (Version 3.1.9.7; Heinrich-Heine-Universität Düsseldorf, Düsseldorf, Germany) indicated an achieved power of 1.00, confirming that the sample size was sufficient to detect the observed effect size (*ƒ*^2^ = 0.18) given the 13 predictors. Among the predictor variables, age, identifying as Asian or Hispanic, prior care partner training, and working in medicine, dentistry, or nursing were significantly associated with lower scores on personhood beliefs. In contrast, SVI, socioeconomic status, identifying as Black, sex, working in behavioral health, public health, or other non-healthcare professions, as well as healthcare experience, were not significant predictors.

Participants identifying as Asian or Hispanic had personhood belief scores that were approximately five points lower than those identifying as White. Individuals with prior formal care partner training scored approximately five points higher than those without such training, suggesting that training may play a meaningful role in strengthening personhood beliefs. Notably, although SVI showed significant group-level differences in the ANOVA analysis, it did not emerge as a significant predictor in the regression model, indicating that its influence may be explained by other overlapping factors when considered simultaneously.

## 4. Discussion

To achieve person- and relationship-centered care, it is essential to foster positive attitudes toward this approach and recognize the innate personhood of individuals receiving support, as emphasized by KAP theory [6]. The ANOVA and regression analyses in this study revealed that age, race, professional discipline, and prior formal care partner training significantly influence personhood beliefs toward PLWD among healthcare professionals and other care partners with small effect size. Although the regression model explained only 15.3% of the variance in personhood beliefs, this modest variance is not unexpected, considering the complexity of human beliefs. Personhood beliefs are shaped by multiple, interconnected factors that may interact with each other through moderating or mediating relationships. The RToP highlights those beliefs about personhood are shaped by a wide range of complex and interrelated influences that are difficult to fully capture [17,18]. While the selection of variables in this study was guided by the RToP, it was also limited to the variables present in the secondary dataset. Consequently, not all dimensions of personhood outlined in the conceptual framework may have been fully represented, which may have contributed to the limited explained variance. For example, variables that capture deeper relational dynamics (e.g., close personal relationships) or broader societal-level influences (e.g., specific societal beliefs or cultural values) were not included, which may have contributed to the relatively low explained variance.

Subscale analyses further revealed that beliefs about PLWD capacity for *psychological engagement* varied by race, age, SVI, professional discipline, prior formal care partner training, and healthcare experience. These findings underscore the importance of raising awareness among healthcare professionals and care partners about how demographic and contextual factors may shape their personhood beliefs toward PLWD. A deeper understanding of these differences is especially critical as dementia prevalence increases with age. In later stages of life, a more adaptive approach may be required to support age-related physical, emotional and cognitive changes, while maintaining dignity and respect for personhood [54].

Regression results indicate that race is a significant predictor of personhood beliefs towards PLWD. Specifically, being Asian is associated with scoring 5.36-points lower and being Hispanic with a 5.63-point lower score, when compared to White respondents. This may reflect cultural contexts in which collective decision-making and viewing the family as the primary unit of care are more common. While such family orientation is often noted in discussions of cultural impact on care, and it offers a potential explanation for these findings, it is important to acknowledge that individuals within any cultural group bring their own values and perspectives that may differ from broader patterns. In these cultural contexts, physicians often consult the family as a whole rather than focusing solely on the individual patient’s preferences [26]. This cultural perspective may explain why Asian and Hispanic healthcare professionals and care partners in our sample reported lower personhood beliefs, especially on statements related to individualized decision-making.

Racial concordance positively influences communication, care satisfaction, and respect [47,48]. Therefore, understanding racial differences in personhood beliefs is crucial for improving care provider/care recipient interactions in the U.S. healthcare system. This need is particularly pronounced given that U.S. population has become more racially and ethnically diverse [29], and the number of PLWD in the U.S. is expected to increase to 11.7 million by 2040 [58]. People’s ability to recall their personal history and experiences may become increasingly limited as dementia progresses [13]. Thus, findings highlight the importance of recognizing cultural differences in personhood beliefs towards PLWD, as they could influence care planning and satisfaction with care. Addressing these disparities is essential for promoting equitable healthcare outcomes for PLWD.

Results show that participants with prior formal care partner training had higher personhood beliefs (4.86 points higher) towards PLWD than those without. Specifically, previously trained participants scored significantly higher across all three subscales, indicating that formal care partner education enhances personhood beliefs. Post-training evaluations from the *Bravo Zulu* training (i.e., the data source for this study) revealed that the training helped reduce racial disparities in personhood beliefs and promoted more positive attitudes [32]. However, although the training narrowed these gaps, it did not fully eliminate identified racial differences, suggesting that education alone may not be sufficient to bridge these disparities [32]. Our findings address the research gap identified by Bejarano et al. [37] regarding the role of healthcare education in improving attitudes toward personhood beliefs. Based on these insights, our results suggest that care partner training may help buffer the influence of contextual factors such as race, age, and professional discipline on personhood beliefs. Care partner training likely provides knowledge and skills that promote self-awareness, allowing participants to critically reflect on their beliefs and reduce the impact of biases shaped by demographic or societal factors. In this way, care partner training could function as a mediator, fostering more positive personhood beliefs by enhancing understanding and empathy toward PLWD. Therefore, we recommend expanding educational initiatives to enhance healthcare professionals’ understanding of personhood and increase awareness of how race and other factors may shape these beliefs. Increased awareness of how race and other factors shape these beliefs can help mitigate biases and improve person- and relationship-centered care.

As individuals progress through different life stages, their beliefs about personhood evolve, shaped by their changing experiences and social contexts. This makes age a critical factor in shaping personhood beliefs. In our sample, age emerged as a marginally statistically significant predictor (*p* = 0.046), with each additional year associated with a 0.07-point decrease in personhood beliefs. Although this effect is very small relative to the overall scale range (20–100), it should be interpreted with caution. However, ANOVA results showed notable differences in personhood beliefs towards PLWD between individuals aged 65 and older (lower level of personhood beliefs) compared to younger age groups. This finding contradicts Newton et al. [35], where it was found that older adults were more likely to feel comfortable around individuals with dementia compared to younger and middle-aged individuals. Several factors may explain this discrepancy. Within the RToP framework, the individual ring includes core attributes such as cognitive abilities, communication, and self-awareness [18]. As individuals age, these attributes interact more dynamically with accumulated life experiences, relationships, and evolving values and beliefs. This integration can create variability in how older adults perceive and express personhood beliefs. Therefore, age may interact with multiple interconnected factors, requiring a more complex model to fully explain its influence on personhood beliefs.

Second, generational effects may play a role. Levinson et al. [36] identified a shift from a patient-centered decision-making approach to a physician-directed style at around age 45, attributing this change to the baby boomer generation reaching their mid-forties, at that time (2005). In contrast, the present study (2020–2024) finds this shift occurring at age 65, aligning with the baby boomer generation nearly all entering elderhood. This suggests that generational aging and societal changes may influence evolving attitudes toward personhood and decision-making in healthcare. Future research with larger, more representative samples and additional variables is needed to clarify whether these patterns represent true conceptual changes in later life or are influenced by confounding factors.

The RToP emphasizes the role of the societal ring, including economic status and societal expectations, in shaping individual perceptions of personhood within a social context [17]. In our sample, participants living in low-medium SVI areas demonstrated higher levels of personhood beliefs towards PLWD compared to those in medium-high SVI areas. At the subscale level, we found significant differences in *psychosocial engagement* between these groups. Additionally, participants who reported receiving public assistance (i.e., lower socioeconomic status) had slightly lower personhood beliefs towards PLWD than those who did not report receiving public assistance. Prior research has shown that individuals living in low-income neighborhoods experience diminished recognition of their personhood [59]. Thus, these findings suggest a potential relationship between *psychosocial engagement* and expectations of person- and relationship-centered care. Considering this, our findings indicate that those residing in lower-medium vulnerability areas or with higher socioeconomic status may have higher personhood beliefs or expectations for a more person- and relationship-centered approach to care compared to individuals in medium-high SVI areas or lower socioeconomic status. Although the effect size of these differences is small, these patterns highlight the importance of raising awareness among professional and non-professional care partners about how socioeconomic factors may influence personhood beliefs.

Consistent with Chochinov et al. [4] our study found differences in personhood beliefs across professional disciplines. Specifically, regression results indicated that participants in ‘Medicine, Dentistry, and Nursing’ discipline had lower personhood belief ratings compared to those in ‘Other health-related professions,’ such as Allied Health. These findings align with the RToP which emphasizes that societal roles, occupational norms, and professional standards shape personhood beliefs [17]. For example, professionals in medicine, dentistry, and nursing often adopt a biomedical and disease-focused approach, emphasizing prevention, diagnosis, and treatment [60]. In contrast, allied health professionals tend to take a more holistic approach to care, incorporating factors such as patients’ functioning and disability, patients’ wants and needs, patients’ ability to participate in care, and patients’ life context [61]. Additionally, within our sample prior healthcare professional experience (defined in the survey as having worked in the healthcare industry as a doctor, nurse, nurse’s aide, or other direct care worker) was not a significant predictor. This finding suggests that specific disciplinary background, rather than general experience in the healthcare field, may play a more influential role in shaping personhood beliefs. Supporting this interpretation, previous research applying the RToP has shown that within intensive care units (ICUs), physicians’ beliefs about personhood were influenced by broader societal factors such as practice standards, work ethics, and professional codes of conduct [18].

Interestingly, no significant differences were found in overall personhood beliefs or subscales based on sex, aligning with Newton et al. [35] who reported no gender differences in attitudes toward dementia or perceptions of personhood. However, our finding contrasts with Bejarano et al. [37] who reported that female healthcare students showed greater support for patient-centered care. These findings suggest that sex influences on personhood beliefs may be context-specific and warrant further investigation, as there are mixed results across various studies regarding the influence of gender [35,37].

Overall, in the context of dementia care, our findings suggest that Asian and Hispanic race, older age, and medicalized professional disciplines were negatively associated with personhood beliefs toward PLWD, while having prior formal care partner training was positively associated with stronger personhood beliefs. Subscale-level analyses further revealed that some demographic and contextual factors influenced specific dimensions of personhood beliefs, even when no differences were observed at the overall scale level. These mixed findings are consistent with the RToP, which emphasizes the interconnectedness of its four rings—innate, individual, relational, and societal—in shaping beliefs about personhood [17,18]. For instance, beliefs related to *agency* (i.e., *PLWD’s* capacity for self-determination) were largely stable across groups, except among those with prior formal care partner training. In contrast, beliefs about *psychosocial engagement* varied across most variables except for sex and socioeconomic status, while *respect for personhood* (i.e., *moral* status and worth of PLWD) differed by race, age, and care partner training.

These patterns support the RToP framework, which posits that personhood beliefs are influenced not only by individual values, but also by broader contextual influences including social, cultural, and professional norms [18]. Prior RToP studies have noted that a supportive environment and accumulated experience can buffer against diminished perceptions of personhood in intensive care settings [18]. In the present study, prior care partner training appears to serve a similar buffering function, reinforcing personhood beliefs among healthcare professionals and other care partners. This finding is consistent with our previous research indicating that culturally responsive, relationship-centered dementia care training can enhance personhood beliefs [32]. Thus, these findings underscore the critical role of care partner training in equipping healthcare professionals and other care partners to recognize and uphold personhood throughout the course of dementia. Personhood is never lost, but it can be overlooked or misunderstood, particularly in the context of cognitive decline, ageism and social bias. Training rooted in person- and relationship-centered principles may help care partners more effectively support the dignity, autonomy, and relational presence of individuals living with dementia, while also fostering self-awareness, cultural humility, and more inclusive care practices in multicultural dementia care settings.

### Limitations

The observed differences in personhood beliefs may be influenced by several factors, including the validity of the instrument when used in an ethnically and racially diverse sample. The PDQ survey tool was originally developed with a Canadian population in mind and did not specifically account for ethnic or racial diversity (P.V. Hunter, 16 February 2024). Although it was later validated for use in a Korean population with cultural adaptations [27], findings showed overall lower attitudes in Korea compared to Canada. While this suggests that the instrument may be applied across cultural contexts, particularly in Asian populations, it has not been validated in a diverse sample like the one in our study. Additionally, the PDQ has not been systematically tested across diverse cultural and professional groups that include both professional (e.g., nurses, physicians, social workers) and non-professional care partners (e.g., family members). While Kim et al. [27] three-factor structure provided the best balance of theoretical grounding and empirical performance, model fit indices were below conventional thresholds. This suggests that certain items may behave differently in our context, and future research needs to evaluate the PDQ’s validity and applicability across diverse cultural and multidisciplinary care settings.

Response bias may also have played a role. Social desirability effects could lead participants from different cultural backgrounds to alter their responses based on perceived cultural expectations. Additionally, self-selection bias may have occurred if individuals with stronger positive beliefs were more likely to register for this type of training.

Although the sample size was reasonably large (*n* = 540), the demographic composition of workshop participants may limit generalizability. For example, 62.5% of the sample had prior professional healthcare experience, which may not reflect the distribution in the general population, particularly among current and future family care partners. Racial group representation also deviated from national U.S. demographics and Nevada’s population, limiting the extent to which racial differences observed in our study can be generalized.

The original secondary dataset had 7.8% missing data, with less than 2% missingness at the individual item level. Because the original scale did not provide specific guidelines for handling missing data, complete-case analysis was used as the primary approach. To confirm the robustness of these findings, a Little’s MCAR test was conducted, which was significant, indicating that data were not missing completely at random. Therefore, we performed a sensitivity analysis using item-level prorating (≤3 missing items) by replacing missing responses with the individual’s average score. Results were consistent with the complete-case model, except that the age effect was no longer significant (*p* = 0.381). Thus, the relationship between age and personhood beliefs should be interpreted cautiously.

Factors considered under the societal domain of the RToP, such as socioeconomic status (SES) and SVI, were measured using relative indicators due to limitations of the secondary dataset. SES was operationalized as a dichotomous indicator (public assistance: yes/no), as individual-level data on income, education, and housing were unavailable. SVI was assigned based on participants’ ZIP codes at the time of the workshop, which may not fully capture their cumulative social and environmental exposures over time. Therefore, the non-significant predictive ability of SES and SVI in the regression analysis should be interpreted cautiously. Future research should use comprehensive individual- and community-level measures to better understand their true impact on personhood beliefs toward PLWD.

While each subgroup analyzed in ANOVA, *t*-tests, and regression had more than 20 participants, the group sizes were unequal (e.g., White = 328, Black = 63, Asian = 53, Hispanic = 49). These disparities may have affected statistical power and the robustness of group-level comparisons.

Finally, while the study was conceptually guided by the RToP, the variables included in the analysis were limited to those available in the secondary dataset. As such, this study could not capture all dimensions outlined in the RToP, potentially omitting key factors such as religious beliefs, self-awareness, ability for cogitation, spiritual beliefs, personal values and cultural identity. Therefore, future research using more comprehensive measures and representative samples are needed to validate and extend these findings.

## 5. Conclusions

Healthcare professionals and other care partners, as well as people receiving care and support, each bring different attitudes, beliefs, and expectations to their care-related encounters. This study provides preliminary evidence that race, professional discipline, and prior care partner training are associated with statistically significant but small differences in healthcare professionals’ and care partners’ attitudes toward PLWD. While SVI and healthcare experience were not significant predictors in the multiple regression analysis, they were associated with significant group differences in ANOVA and *t*-tests with small effect size. This suggests that these factors have a minimal potential influence on beliefs about the psychological and social capacities of PLWD. Because these findings are derived from secondary data and a non-generalizable sample, they should be interpreted with caution and do not establish causal relationships. Nonetheless, this study provides a foundation for future research using representative samples and expanded measures to better understand how demographic and contextual factors shape personhood beliefs and to guide the development of strategies for more person-centered dementia care. Increasing awareness of these differences can support the development of greater self-awareness among healthcare professionals and other care partners, helping them recognize how their own experiences and beliefs may shape care-related decisions. For example, our findings indicate that White and Black care partners reported more positive personhood beliefs compared to their Asian and Hispanic counterparts. Additionally, although age was only a marginal significant predictor in regression analysis, individuals aged 65 years and older exhibited lower personhood beliefs than their younger counterparts. Recognizing these variations may help healthcare professionals and other care partners overcome potential barriers, align care approaches with the expectations and values of those receiving care, and ultimately improve quality of care, satisfaction with care, and the well-being of PLWD. These findings indicate clear program and policy implications, suggesting a need for future dementia care training to consider incorporating elements related to cultural sensitivity and acknowledging disciplinary differences in dementia care roles. Thus, future person- and relationship-centered dementia care programs may consider incorporating elements to support learning outcomes for participants with varied personal and professional backgrounds. Ultimately, recognizing the need for such elements could enhance participant training outcomes and improve quality of care.

## Figures and Tables

**Table 1 ijerph-22-01491-t001:** Characteristics of Study Participants (*n* = 540).

Characteristics	Frequency (*n*)	Percent (%)
**Sex**		
Male	68	12.6
Female	472	87.4
**Ethnicity/Race**		
American Indian or Alaska Native	2	0.4
Asian	53	9.8
Black or African American	63	11.7
Hispanic	49	9.1
Native Hawaiian or other Pacific Islander	3	0.6
White	328	60.7
Multiple Race	33	6.1
Other	9	1.7
**Age Groups**		
18 to 34 years	104	19.3
35 to 44 years	151	28.0
45 to 54 years	124	23.0
55 to 64 years	100	18.5
65 years and over	60	11.1
Missing/not reported	1	0.2
Experience as a Healthcare Professional	338	62.6
DO NOT have Experience as a Healthcare Professional	202	37.4
**Professional Discipline**		
Behavioral Health	161	29.8
Medicine, Dentistry and Nursing	79	14.6
Public Health	22	4.1
Other Health Related Professions	78	14.4
Other Non-Healthcare Related Professions	93	17.2
Profession missing/not reported	107	19.8
**Social Vulnerability Level Based on County of Residence**		
Low	65	12.0
Low-Medium	23	4.3
Medium-High	324	60.0
High	126	23.3
Missing (due to missing ZIP code or error in ZIP code)	2	0.4
**Socioeconomic Status**		
Family DO NOT Receive Public Assistance	466	86.3
Family Receive Public Assistance	73	13.5
Socioeconomic status missing	1	0.2
Participants with Prior Formal Care Partner * Training	372	68.9
Participants without Prior Formal Care Partner * Training	168	31.1

* While survey data was collected using the term ‘caregiver’ within validated tools and demographic questions, the authors’ preferred terminology is ‘care partner’ and thus that term is used throughout the text and tables in this manuscript. The demographic composition presented here excludes outliers.

**Table 2 ijerph-22-01491-t002:** Comparison of Mean Total PDQ Scores Across Different Demographic and Socioeconomic Groups.

Measures	Results	Post Hoc Comparisons	Groups	*n*	*M*	*SD*
Race ^a,d^	*F*(3,489) = 7.02, *p* < 0.001,	Asian vs. White *p* = 0.008	White	328	88.81	8.96
	*η*^2^ = 0.04 (small effect)	Hispanic vs. White *p* = 0.002	Black	63	88.49	9.74
		Hispanic vs. Black *p* = 0.033	Asian	53	84.49	9.24
			Hispanic	49	83.76	9.19
SVI ^a,d^	*F*(3,534) = 3.34, *p* = 0.019,	Medium-High vs. Low Medium *p* = 0.018	Low	65	88.92	9.92
	*η*^2^ = 0.02 (small effect)		Low-Medium	23	92.78	7.97
			Medium-High	324	86.95	9.16
			High	126	87.59	9.56
Age ^a,d^	*F*(4,534) = 4.48, *p* = 0.001,	65 over vs. 18 to 34 *p* = 0.028	18 to 34 years	104	87.48	9.07
	*η*^2^ = 0.03 (small effect)	65 over vs. 35 to 44 *p* = 0.004	35 to 44 years	151	88.07	9.95
		65 over vs. 45 to 54 *p* = 0.005	45 to 54 years	124	88.15	9.02
		65 over vs. 55 to 64 *p* < 0.001	55 to 64 years	100	89.12	8.22
			65 years and over	60	83.07	9.83
Professional Discipline ^a,d^	*F*(4,428) = 2.98, *p* = 0.019,	Other health professional vs. Behavioral health *p* = 0.017	Behavioral Health	161	87.07	9.3
	*η*^2^ = 0.03 (small effect)		Medicine, Dentistry and Nursing	79	87.51	9.48
			Public Health	22	90.73	8.34
			Other Health Professions	78	91.03	9.17
			Other Non-Healthcare Professions	93	88	9.06
Socioeconomic Status ^b,c^	*t*(105.52) = −1.27, *p* = 0.208	n/a	Received public assistance	73	86.47	8.15
			Not received public assistance	466	87.8	9.55
Sex ^b,d^	*t*(538) = −1.31, *p* = 0.684	n/a	Male	68	87.19	9.37
			Female	472	87.69	9.38
Formal Care Partner Training ^b,d^	*t*(538) = 4.828, *p* < 0.001	n/a	Yes	372	88.91	8.89
	Cohen’s d = 0.45 (small effect)		No	168	84.78	9.8
Experience as a Healthcare Professional ^b,d^	*t*(538) = 1.741 *p* = 0.082	n/a	Yes	338	88.17	9.16
			No	202	86.72	9.67

Note: ^a^ One-way ANOVA test. ^b^ independent *t*-test. ^c^ equal variance not assumed. ^d^ homogeneity of variance was met as assessed by Levene’s test.

**Table 3 ijerph-22-01491-t003:** Comparison of Mean Subscale Level PDQ Scores Across Different Demographic and Socioeconomic Groups.

Measures	Subscales		
	Agency	Respect for Personhood	Psychosocial Engagement
Race	*F*(3,489) = 1.62, *p =* 0.183 ^a^	*F*(3,449) = 6.66, *p* < 0.001, *η*^2^ = 0.04 ^b^ (small effect) Asian vs. White *p* = 0.017 Hispanic vs. White *p* = 0.001	*F*(3,489) = 5.99, *p* < 0.001, *η*^2^ = 0.03 ^b^ (small effect) Asian vs. White *p* = 0.012 Hispanic vs. White *p* = 0.010 Hispanic vs. Black *p* = 0.043
SVI	*F*(3,534) = 1.10, *p* = 0.350 ^a^	*F*(3,534) = 1.61, *p* = 0.185 ^a^	*F*(3,534) = 4.45, *p* = 0.004, *η*^2^ = 0.03 ^b^ (small effect) Medium-High vs. Low Medium *p* = 0.009
Age	*F*(4,534) = 1.58, *p =* 0.179 ^a^	*F*(4,534) = 3.85, *p* = 0.004, *η*^2^ = 0.03 ^b^ (small effect) 65 over vs. 18 to 34 *p* = 0.011 65 over vs. 35 to 44 *p* = 0.024 65 over vs. 45 to 54 *p* = 0.009 65 over vs. 55 to 64 *p* = 0.003	*F*(4,534) = 4.54, *p* = 0.001, *η*^2^ = 0.03 ^b^ (small effect) 65 over vs. 18 to 34 *p* = 0.047 65 over vs. 35 to 44 *p* = 0.007 65 over vs. 45 to 54 *p* = 0.009 65 over vs. 55 to 64 *p* < 0.001
Professional Discipline	*F*(4,428) = 3.39, *p* = 0.010, *η*^2^ = 0.03 ^b^ Other health professional vs. Behavioral health *p* = 0.004	*F*(4,428) = 1.04, *p* = 0.389 ^a^	*F*(4,428) = 4.02, *p* = 0.003, *η*^2^ = 0.04 ^b^ (small effect) Other health professions vs. Behavioral health *p* = 0.005 Other health professions vs. Other non-health professions *p* = 0.013
Socioeconomic Status	*t*(113.13) = −0.771, *p =* 0.442 ^c^	*t*(537) = −0.02, *p =* 0.984	*t*(537) = −1.91, *p =* 0.056
Sex	*t*(538) = 0.51, *p =* 0.612	*t*(76.71) = −1.35, *p =* 0.180 ^c^	*t*(538) = 0.77, *p =* 0.707
Formal Care Partner Training	*t*(538) = 3.52, *p* < 0.001 Cohen’s d = 0.33 (small effect)	*t*(261.50) = 3.44, *p* < 0.001 ^c^ Cohen’s d = 0.35 (small effect)	*t*(538) = 4.59, *p* < 0.001 Cohen’s d = 0.42 (small effect)
Experience as a Healthcare Professional	*t*(538) = −0.17, *p =* 0.863	*t*(538) = 1.13, *p =* 0.260	*t*(538) = 2.67, *p =* 0.008 Cohen’s d = 0.24 (small effect)

Note: ^a^ Post Hoc comparison not significant. ^b^ Post Hoc comparison significant. ^c^ equal variance not assumed.

**Table 4 ijerph-22-01491-t004:** Multiple Regression Analysis of Predictors for Personhood Beliefs Towards PLWD.

Variable	*B*	*SE*	*t*	*p*	95% CI		
					LL	UL	VIF
Race ^a^							
Black	−1.71	1.31	−1.30	0.193	−4.29	0.87	1.09
Asian	−5.36	1.38	−3.88	<0.001	−8.07	−2.64	1.07
Hispanic	−5.63	1.51	−3.74	<0.001	−8.59	−2.67	1.12
SVI	−1.80	1.87	−0.96	0.337	−5.47	1.88	1.04
Age	−0.07	0.04	−2.00	0.046	−0.15	0.00	1.15
Professional Discipline ^b^							
Behavioral Health	−2.36	1.28	−1.84	0.066	−4.88	0.16	2.07
Medicine, Dentistry and Nursing	−3.72	1.42	−2.62	0.009	−6.52	−0.93	1.72
Public Health	0.54	2.35	0.23	0.819	−4.08	5.15	1.30
Other Non-Healthcare Related Profession	−1.01	1.47	−0.69	0.490	−3.90	1.87	1.96
Socioeconomic Status ^c^	−0.90	1.26	−0.71	0.478	−3.38	1.58	1.04
Sex ^d^	−0.61	1.27	−0.48	0.630	−3.10	1.88	1.02
Formal Care Partner Training ^c^	4.86	0.99	4.92	<0.001	2.92	6.81	1.16
Experience as a Healthcare Professional ^c^	1.43	1.03	1.38	0.169	−0.61	3.46	1.29

Note: CI confidence interval; LL lower limit, UL upper limit. ^a^ race "White” benchmark category. ^b^ Other health related professions as benchmark category. ^c^ 0 = no, 1 = yes. ^d^ 0 = female, 1 = male.

## Data Availability

The raw data supporting the conclusions of this article will be made available by the authors on request.

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
