# Peer review of "Personhood Beliefs in Dementia Care: Influences of Race, Socioeconomic Factors, and Social Vulnerability"

_ijerph, 2025, doi:10.3390/ijerph22101491_

Round 1

Reviewer 1 Report

Comments and Suggestions for Authors

Dear Authors,

This manuscript addresses a timely and socially relevant topic concerning personhood beliefs in dementia care, framed through the Ring Theory of Personhood (RToP). The use of a large sample and the integration of demographic and contextual variables provide a strong foundation. However, several areas of the manuscript require clarification, conceptual integration, or deeper interpretation before it is suitable for publication.

Please find below my detailed comments, organised by manuscript section:

ABSTRACT

Comment 1 (Line 25)

There is a potential inconsistency in how SVI and healthcare experience are presented. While described as not significant predictors in the regression model, they are reported as relevant in subscale analyses.

The authors should clarify this point in the abstract to avoid misleading readers.

INTRODUCTION

Comment 2 (Lines 33–49)

The introductory paragraphs define person-centered and relationship-centered care with overlapping content, leading to redundancy.

The authors should streamline this section into a more concise and integrated explanation of both concepts.

Comment 3 (Lines 91–95)

The research questions are presented in interrogative form, which reduces the clarity of the study objectives.

The authors should rephrase the aims as explicit objectives to improve precision (e.g., "This study aimed to…").

CONCEPTUAL FRAMEWORK

Comment 4 (Lines 149–158)

Although the SVI is a core variable in the analysis, its conceptual alignment with the "societal ring" of the RToP is not established.

The authors should explain how community-level vulnerability indicators fit within the RToP framework.

Comment 5 (Lines 171–181)

The manuscript conflates general healthcare experience with formal dementia care training, which are conceptually distinct.

The authors should clarify this distinction in both the conceptual and analytical parts of the manuscript.

METHODS

Comment 6 (Lines 227–234)

The justification for modifying the PDQ Likert scale from 7 to 5 points is not supported by psychometric evidence.

The authors should add a statement noting that high internal consistency (α = 0.90) in this study supports the validity of the modified scale.

Comment 7 (Line 254)

Two outliers were removed from the sample, but no information is given on whether this impacted the demographic composition.

The authors should confirm whether the removal of outliers altered the representativeness of key subgroups.

Comment 8 (Lines 294–296)

Age and SVI are treated as continuous variables in regression models but were introduced categorically earlier. The rationale for this shift is not explained.

The authors should justify this analytical choice and ensure consistency across sections.

RESULTS

Comment 9 (Lines 313–315)

The decline in personhood belief scores among participants aged 65+ is not discussed, despite its potential importance.

The authors should consider whether this is due to sample size limitations, generational effects, or conceptual changes in older adulthood.

Comment 10 (Lines 338–340)

The finding that individuals in medicine, dentistry, and nursing scored lower on personhood beliefs is underexplored.

The authors should propose possible interpretations—e.g., effects of biomedical training or differing care philosophies.

DISCUSSION

Comment 11 (Lines 367–368)

Several parts of the discussion restate findings without further interpretation or theoretical integration.

The authors should enhance the discussion by exploring how demographic factors may interact (e.g., as moderators or mediators) within the RToP model.

Comment 12 (Lines 397–403)

Although prior care partner training emerges as a strong predictor, its broader impact is not theorised.

The authors should suggest whether training might mediate or buffer the influence of contextual factors on personhood beliefs.

Comment 13 (Lines 457–462)

The discussion of sex and gender is inconclusive and somewhat inconsistent with prior research.

The authors should acknowledge the mixed evidence in the literature and advocate for more nuanced, gender-informed studies in future research.

LIMITATIONS

Comment 14 (Lines 497–503)

Potential social desirability bias is not addressed, despite the use of self-reported data in a training context.

The authors should add a note on the risk of socially desirable responding as a limitation of the study design.

Comment 15 (Lines 515–518)

The manuscript briefly notes that some RToP domains were not captured, but this is underdeveloped.

The authors should emphasise the limitations posed using secondary data and the absence of measures related to spirituality, personal values, and cultural identity.

CONCLUSIONS

Comment 16 (Lines 531–539)

The conclusions summarise findings but do not offer practical or policy-relevant implications.

The authors should add a closing statement linking results to the design of culturally sensitive and role-specific training interventions.

Best regards

Author Response

Please find attached the revised manuscript along with our detailed responses to the reviewers’ comments. We have made all requested changes and incorporated additional analyses to strengthen the manuscript, following the feedback provided by all three reviewers.

In particular, we conducted a Confirmatory Factor Analysis (CFA) as recommended, and the results have been included in the Methods section. We are happy to share the full CFA output upon request.

For your reference, we have also attached a document outlining our responses to each reviewer comment, as well as a tracked-changes version of the manuscript highlighting all modifications made.

Thank you for the opportunity to revise and improve our work. We greatly appreciate the reviewers’ thoughtful feedback and look forward to your consideration.

Reviewer 2 Report

Comments and Suggestions for Authors

STRENGTHS

  1. Timely and Relevant Topic: The manuscript addresses an important and underexplored issue in dementia care—how personhood beliefs are influenced by socio-demographic factors such as race, socioeconomic status, and professional discipline.
  2. Theoretical Framework: The integration of the Ring Theory of Personhood (RToP) provides a coherent conceptual structure.
  3. Robust Methodology: The use of both bivariate and multivariate analyses enhances the credibility of the findings.
  4. Sample Size and Diversity: A relatively large and diverse sample (n = 540), with inclusion of both professional and non-professional care partners, enhances the generalizability and richness of the results.
  5. Instrument Adaptation and Reliability: The adaptation of the PDQ to a 5-point scale is well-justified and supported by literature.

AREAS FOR IMPROVEMENT

  1. Instrument Validity in Diverse Populations: The authors acknowledge that the PDQ was not originally validated in ethnically diverse populations. However, further critical reflection on how this may have biased findings would strengthen transparency.
  2. Conceptual Tension on Age: The finding that older adults (>65) had lower personhood beliefs contradicts previous research cited. While the authors speculate generational shifts, a more detailed discussion or theoretical anchoring is warranted to resolve this tension.
  3. Limited Variance Explained: The regression model accounts for only 15.3% of the variance in personhood beliefs. While the authors recognize this and link it to limitations of secondary data, a more explicit discussion on what additional variables might be important would be beneficial.
  4. Clarity on 'Medicalized Professions': The manuscript reports lower personhood beliefs among those in "medicine, dentistry, and nursing" compared to “other health professions.” The rationale behind this grouping should be more clearly explained.
  5. Limitations Section: the limitations section could benefit from a clearer differentiation between methodological limitations.

Author Response

(The authors gave the same response as above.)

Reviewer 3 Report

Comments and Suggestions for Authors

1. Originality and Relevance

The study addresses a current and highly relevant question: how personhood beliefs in dementia care are shaped by sociodemographic, disciplinary, and contextual factors.

The use of the Ring Theory of Personhood (RToP) as a theoretical framework is appropriate and innovative, since it integrates individual, relational, and societal dimensions.

However, the scientific gap is not fully articulated: the manuscript presents literature reviews on age, race, comunity-level social vulnerability (SVI), and professional discipline, but does not systematically demonstrate what remains unknown. A clearer critical review of the state of the art is recommended, explicitly identifying:

  • how many and which previous studies addressed factors such as SVI and personhood beliefs;
  • whether comparative literature exists in multicultural contexts;
  • to what extent this study is the first to integrate these variables together.

2. Methodology

2.1 Population and Sample

  • The use of secondary data from the Bravo Zulu program is valid, but raises representativeness issues:
    • 62.5% of participants had healthcare experience, biasing the results toward a population more knowledgeable about care.
    • 80% resided in Nevada, limiting generalizability to the U.S. states context.
  • The absence of a priori sample size/power calculation should be acknowledged. Although n=540 is adequate for multiple regression, the lack of expected effect size estimation undermines the assessment of statistical robustness.

2.2 Measurement Instrument (PDQ)

The Personhood in Dementia Questionnaire (PDQ) was altered:

  • Substitution of “resident with dementia” with “people living with dementia.”
  • Reduction of the Likert scale from 7 to 5 points.

Although the authors cite literature supporting the equivalence between 5- and 7-point scales, they do not present their own psychometric analysis to confirm the consistency of this adaptation in this sample. It is recommended to:

  • conduct a Confirmatory Factor Analysis (CFA) to verify whether the 3-subscale structure holds in the modified version;
  • present construct validity indicators (CFI, TLI, RMSEA), in addition to Cronbach’s alpha.

2.3 Definition of Independent Variables

  • Socioeconomic status was reduced to a dichotomous indicator (“receives or does not receive public assistance”). This simplification may overlook critical nuances of income, education, and housing conditions.
  • SVI was assigned at the county level based on ZIP code. This generates a risk of ecological fallacy, as it does not ensure that all individuals share the same social vulnerability of the geographic unit. An alternative would be to include individual-level vulnerability variables.
  • The definition of healthcare professional experience is broad (physician, nurse, aide), but lacks stratification. Group heterogeneity may dilute disciplinary effects.

2.4 Statistical Procedures

  • Exclusion of missing data: Listwise deletion may bias results if data are not missing completely at random (MCAR). No test was reported (e.g., Little’s MCAR test). It is recommended to evaluate missing data patterns and consider multiple imputation.
  • ANOVA and t-tests: Application is appropriate, but it was not reported whether assumptions of normality and homoscedasticity (Kolmogorov-Smirnov, Levene’s test) were verified.
  • Correction for multiple comparisons: Not applied (e.g., Bonferroni or FDR). Given the number of analyses, there is a high risk of Type I errors.
  • Multiple regression:
    • The model explains only 15.3% of the variance (R²). The low explanatory capacity must be explicitly acknowledged.
    • No analysis of multicollinearity (VIF, tolerance) was presented, which could compromise coefficients.
    • Categorical variables with low frequency (e.g., race n=2 for American Indian) were excluded, but this should be discussed as a limitation of underrepresentation.

3. Results

The presentation is clear and tables are detailed. However:

  • Some mean differences are statistically significant but of small practical effect (η² < 0.04, d < 0.5). This should be highlighted to avoid overstated interpretations.
  • The discussion on age ≥65 being associated with lower personhood beliefs should be contextualized as an exploratory trend, given intergenerational variability and possible confounders (e.g., physical health, health literacy not assessed).

4. Discussion

  • The discussion is rich, but in some points there are risky cultural extrapolations (e.g., attributing Asian and Hispanic score differences to family decision-making practices). Such hypotheses should be presented as speculative, not conclusive.
  • The link to theory (RToP, KAP) is well established, but could be more critical: the study did not directly assess attitudes or practices, only beliefs, therefore the KAP framework may be overstretched.
  • The role of prior care partner training is correctly highlighted, but the authors do not consider the possibility of self-selection bias (individuals predisposed to positive beliefs may be the ones more likely to seek training).

5. Limitations

The limitations are acknowledged, but could be expanded:

  • Transcultural validity of the PDQ — not validated in this diverse population.
  • Representativeness — sample biased toward healthcare professionals and Nevada residents.
  • Ecological fallacy in the use of SVI.
  • Exclusion of relevant contextual variables (religion, personal experience with dementia, family support).
  • Cross-sectional design — does not allow causal inference or temporal evolution of beliefs.

6. Conclusions and Implications

  • Conclusions should be more moderated. Instead of stating that “race, age, and professional discipline predict beliefs,” it would be more accurate to say that these factors are associated with statistically significant but small differences.
  • Practical implications (culturally responsive training, professional self-awareness) are valid, but should be supported by a more critical discussion of the strength of evidence.

Necessary changes:

  1. Strengthen the theoretical framework and explicitly state the scientific gap.
  2. Provide psychometric justification for the PDQ adaptation (ideally with CFA).
  3. Revise statistical strategy: test assumptions, apply corrections for multiple comparisons, report VIF.
  4. Present results with emphasis on effect size rather than statistical significance alone.
  5. Make the discussion more cautious, avoiding unsupported cultural extrapolations.
  6. Reinforce limitations, especially transcultural validity of the instrument and sample representativeness.

Author Response

(The authors gave the same response as above.)

Round 2

Reviewer 1 Report

Comments and Suggestions for Authors

Dear Authors,

After reviewing the revised version of your manuscript, I confirm that the comments raised during the first round of review have been addressed. The passages highlighted in yellow correspond to the modifications you introduced in response to the reviewer’s feedback. These include clarifications in the abstract, a more concise and integrated introduction, strengthened alignment of the conceptual framework, improved methodological justification, expanded interpretation of results, and a more detailed discussion of limitations and conclusions.

Overall, the revisions have enhanced the clarity, theoretical consistency, and practical relevance of the paper. The manuscript is now substantially improved and ready for further editorial consideration.

Best regards 

The reviewer

Author Response

Thank you for the feedback.